# Decontamination of *Geobacillus Stearothermophilus* using the Arca Aerosolized Hydrogen Peroxide decontamination system

**Loren Benjamin Mead**[1,2]*, **Tanner Mathison**[1], **Garry Osborne**[1], **Anne Marie Richards**[1]

**1** Abaton, Board of Directors, Washington, D.C, United States of America, **2** Department of Emergency Medicine, University of Texas Health Science Center at Houston, Houston, TX, United States of America

* lmead@abaton.care

**Data Availability Statement:** The testing results have been published on Dryad at https://datadryad.org/stash/share/prBqIlz7DDystlxHEBmpo6TWigdvi8OyjTi9oViQJqY

## Abstract

### Introduction

In response to the limited supply of personal protective equipment during the pandemic caused by SARS-CoV-2, recent studies demonstrate that gaseous $H_2O_2$ is an effective decontaminant of N95 filtering facepiece respirators to enable reuse of these items in a clinical setting. This paper evaluates the efficacy of the Arca Aerosolized Hydrogen Peroxide Decontamination System (Arca), a novel aerosolized $H_2O_2$ decontamination system, using biologic indicator testing.

### Materials and methods

The Arca produces and circulates $H_2O_2$ aerosol inside of a sealed stainless steel chamber. The Arca's decontamination efficacy was evaluated in 8 decontamination trials with 2 $H_2O_2$ concentrations (3% and 12%) and 4 decontamination cycle durations (45, 60, 90, and 120 minutes). Efficacy was evaluated by testing: 1) the concentration in parts per million (ppm) of $H_2O_2$ produced inside the chamber and the concentration in ppm of $H_2O_2$ vented from the chamber, and 2) the decontamination of Mesa Biologic Indicator filter strips (BI) inoculated with *Geobacillus Stearothermophilus*. Control tests were conducted by submerging BI strips in 3mL of 3% and 12% $H_2O_2$ for 120 minutes (negative controls) and by not exposing one BI strip to $H_2O_2$ (positive control).

### Results

Greater than 5000 ppm of $H_2O_2$ was detected on the concentration strips inside the chamber for each of the eight decontamination trials. No vented $H_2O_2$ was detected on the external concentration strips after any decontamination trial. No growth was observed for any of the negative controls after seven days. The positive control was positive for growth.

### Conclusion

The Arca Aerosolized Hydrogen Peroxide Decontamination System is effective at decontaminating bacterial *G. Stearothermophilus* at a cycle time of 45 minutes utilizing 6mL of 3% $H_2O_2$ solution.

**Funding:** The Booz Allen Foundation Innovation Fund contributed $25,000 to Abaton, a non profit corporation for the development and testing of the Arca device, including the research published in this paper. The Booz Allen Foundation is a separate legal entity from Booz Allen Hamilton and it exists as a 501(c)3 with the mission "To be a convening force for the social sector, bringing together thought leaders and diverse perspectives to use innovation and technology advancements to help solve challenging social issues and build community resilience from the ground up". No authors directly received funding from any source, including the Innovation Fund. The Innovation Fund website can be reached at https://boozallenfoundation.org/innovationfund/. The funders had no role in study design, data collection and analysis, decision to publish, or preparation of the manuscript.

**Competing interests:** The authors of this publication are members of Abaton's Board of Directors, which could be perceived as a competing interest. However, the authors do not receive a salary or any gifts-in-kind for their service on the Board, and Abaton is a 501(c)3 non profit. The Arca device design has not, nor ever will be submitted for patent. The intention of the authors is to disseminate the design as an open source document for replication in locations where it might be of assistance. The authors further declare that they in no way financially or otherwise benefitted from the grant funding and personal donations made to Abaton. This does not alter our adherence to PLOS ONE policies on sharing data and materials.

## Introduction

The SARS-CoV-2 coronavirus (COVID-19) pandemic has caused a worldwide shortage in personal protective equipment (PPE) including N95 filtering facepiece respirator (FFR) and surgical masks. Both mask types have been shown to dramatically reduce the rate of infection by airborne viruses such as influenza, but the masks are designed for single use before disposal [1]. Due to a shortage of FFRs and the critical role FFRs play in protecting health care workers at the outset of the COVID-19 pandemic, the Center for Disease Control recommended limited reuse of decontaminated FFRs to extend supplies of PPE [1,2]. The Occupational Safety and Health Administration has similarly produced guidelines for the limited reuse of FFRs [3].

Aerosolized or vaporized hydrogen peroxide ($H_2O_2$) is an established reactant for decontaminating surfaces inoculated with resistant bacterial and viral pathogens [4]. Recent studies demonstrate that $H_2O_2$ delivered in an aerosol or vapor as low as 500 ppm has been shown to achieve 1000 times reductions in viral organism activity on inoculated FFRs while maintaining adequate mask fitment through multiple decontamination cycles [5–8]. Previous work has demonstrated that scalable, proof-of-concept $H_2O_2$ decontamination systems could achieve adequate minimum concentrations reported in the literature to eradicate Coronavirus and other pathogens [9,10].

To mitigate infection risk with reuse of FFRs, the Food and Drug Administration granted emergency use authorizations (EUAs) to several commercially available hydrogen peroxide FFR decontamination procedures [11]. However, the commercially available methods of FFR decontamination are costly and limited in availability [12–14]. The $H_2O_2$ decontamination methods which received an EUA include those listed in Table 1. The number of maximum decontamination cycles per FFR varies by method with a range of two to 20 cycles [15]. The price per institution for these systems can exceed $50,000 (Table 1). Low-resource institutions are often unable to access these services and purchase adequate supplies of PPE [16]. Therefore, the development of a smaller-scale, less-costly decontamination devices is warranted.

This study seeks to evaluate the efficacy of a low-cost design engineered for use in low-resource settings to expand the supply of decontaminated PPE for safe re-use by frontline workers. Our objectives were to test the device's efficacy at different $H_2O_2$ concentrations and Arca device cycle durations by evaluating: 1) the concentration in parts per million (ppm) of $H_2O_2$ produced inside the chamber and the concentration in ppm of $H_2O_2$ vented from the chamber, and 2) the decontamination of Mesa Biologic Indicator filter strips (BI) inoculated with $10^6$ *Geobacillus Stearothermophilus*.

**Table 1. Value Comparison of $H_2O_2$ decontamination systems with EUAs.**

| Decontamination System | Estimated Annual Cost to Institution | Maximum Decontamination Cycles per FFR |
|---|---|---|
| Bioquell Technology System | $53,000 [17] | 4 |
| Battelle Critical Care Decontamination System | $0[a] | 20 |
| STERRAD Sterilization System | $149,000 [18] | 2 |
| Sterilucent HC 80TT Vaporized Hydrogen Peroxide Sterilizer | Not Reported | 10 |
| Stryker STERIZONE VP4 Sterilizer for N95 Respirator Decontamination | Not Reported | 2 |
| Stryker Sustainability Solutions VHP N95 Respirator Decontamination System | Not Reported | 3 |
| Duke Decontamination System | Not Reported | 10 |
| Technical Safety Services (TSS) 20-CS Decontamination System | Not Reported | 20 |
| Michigan State University Decontamination System | Not Reported | 3 |
| Roxby Development Zoe-Ann Decontamination System | Not Reported | 4 |

[a]Battelle was awarded a federal contract for subsidized N95 decontamination with costs upwards of $1 million per system [19].

## Materials and methods

The Arca Aerosolized Hydrogen Peroxide Decontamination System (Arca) produces $H_2O_2$ inside of a sealed stainless steel chamber and circulates the aerosol with four CG IP67 personal computing fans. The Arca uses a Venturi tube design to generate an aerosol: an air compressor forces high velocity air through a tube where the cross-sectional area is reduced with the result of lowering the air pressure (Fig 1). The design lowers the pressure to below 2300 Pa, the vaporization pressure of 10% $H_2O_2$ in water solution [20]. The $H_2O_2$ is fed to the low pressure opening of the tube, where the low pressure induces partially vaporization, forming small droplets of solution (i.e. an aerosol). A catch is included at the low pressure point to trap large droplets and recirculate them back to the low-pressure area. FFRs are placed on a wire mesh rack inside the chamber and exposed to the $H_2O_2$ for decontamination (Fig 2). Circulated $H_2O_2$ aerosol is removed from the chamber interior using a Speedair 4ZL07 condenser unit to avoid operator exposure when removing decontaminated FFRs (Fig 3). The device's manufacturing and component costs totaled less than $2,000.

The Arca's decontamination efficacy was evaluated by a testing protocol exposing Mesa-Strip BI filter strips inoculated with $10^6$ *G. Stearothermophilus* [22]. We performed a total of 8 trials varying the $H_2O_2$ concentration (3% and 12%) and decontamination cycle duration (45, 60, 90, and 120 minutes). Decontamination efficacy was evaluated by $H_2O_2$ concentration and biologic testing within the device chamber and venting system.

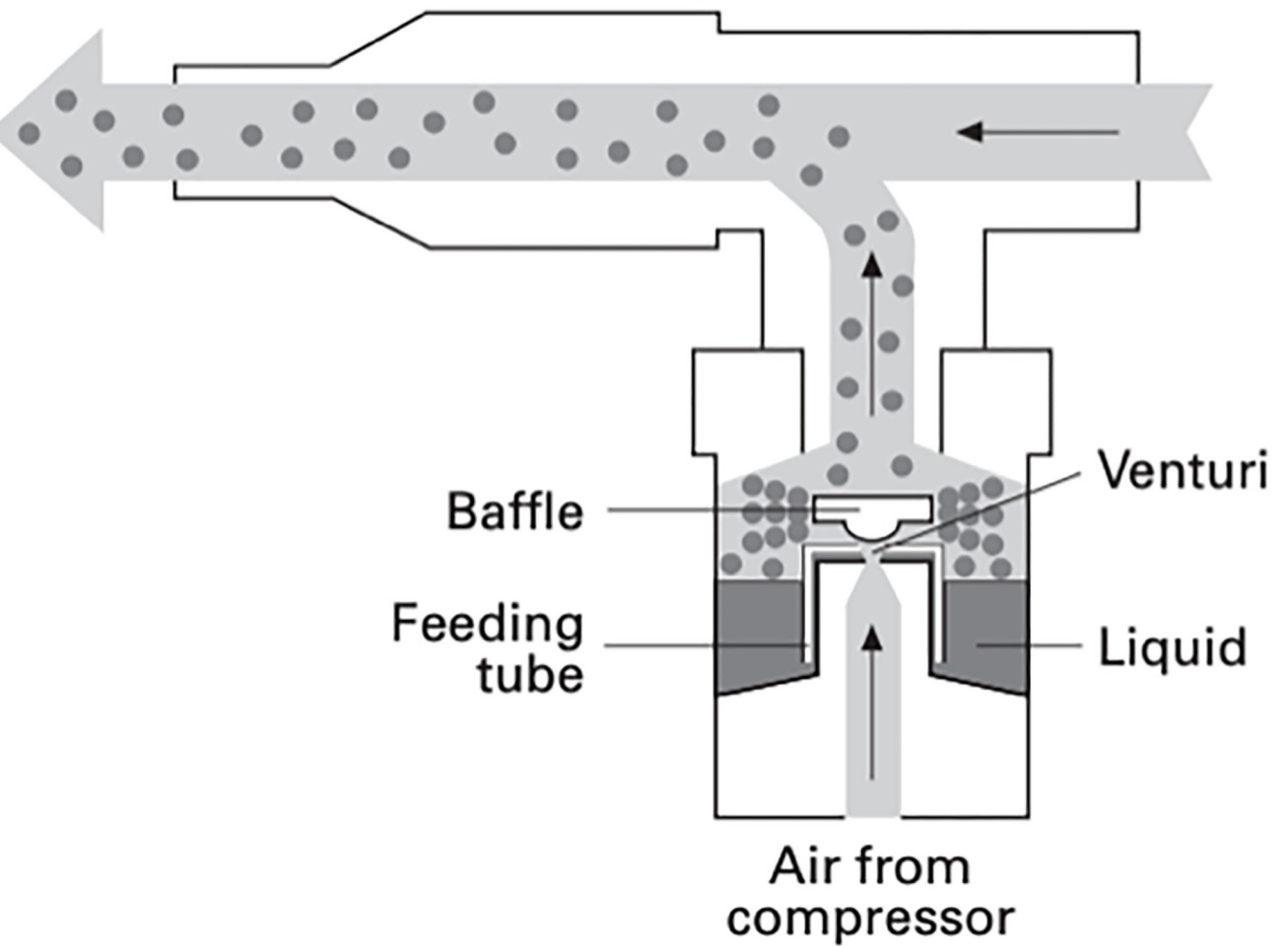

**Fig 1. Nebulizer Venturi tube design** [21].

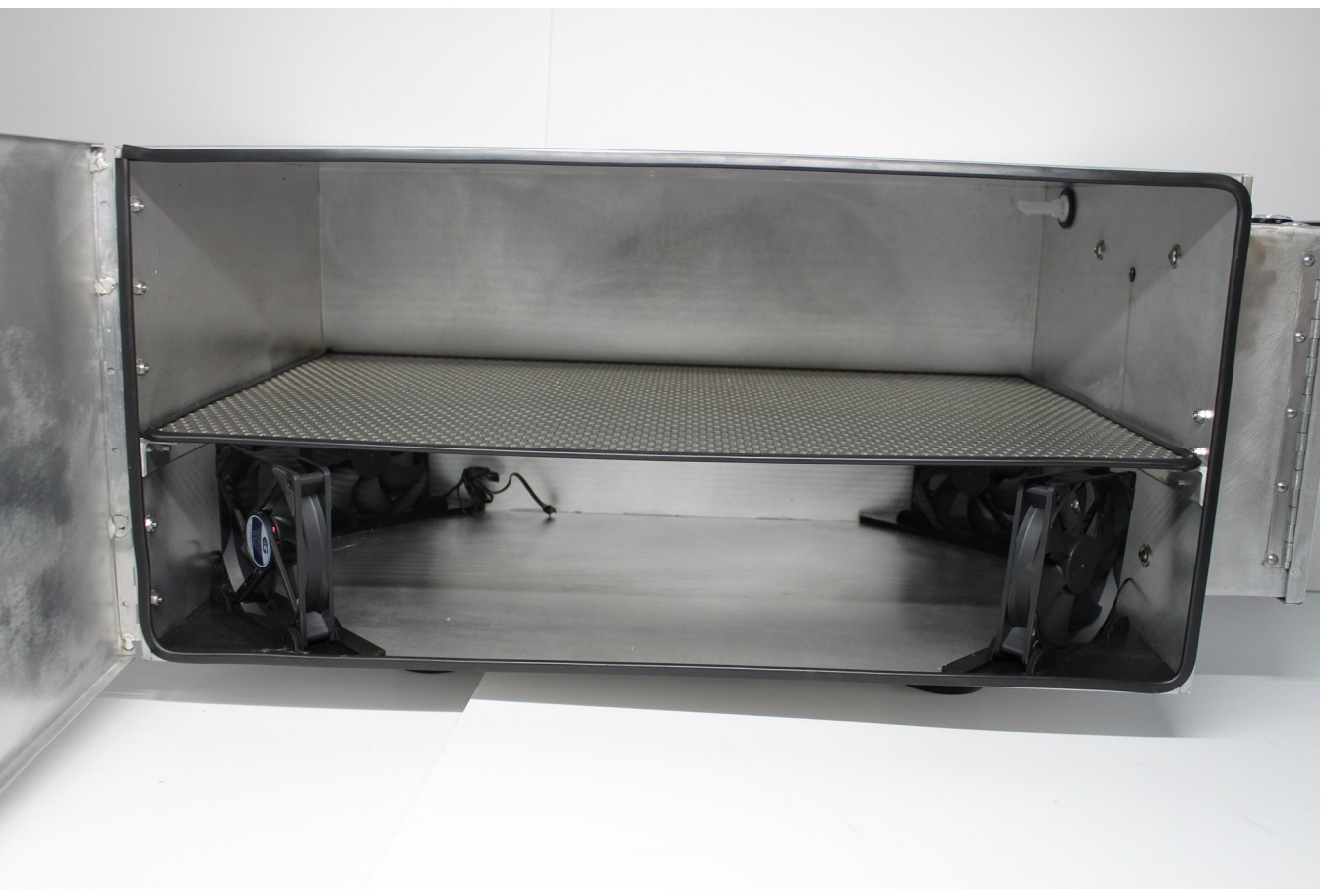

**Fig 2. Arca interior demonstrating mesh rack, circulating fans, and Venturi tube nozzle.**

Concentration testing was performed using Bartovartion Very High Level Peroxide Test Strips, which change color to indicate exposure to $H_2O_2$ at concentrations from 0 to 5000 ppm [23]. For each trial, one BI was secured with Scotch Magic$^{TM}$ Tape to an FFR located centrally inside the chamber. An additional concentration strip was placed on the exterior of the door and evaluated for vented $H_2O_2$ from the chamber after each cycle was completed and the door to the chamber opened. All cycles were performed in a well-ventilated area.

Additionally, one BI was submerged in 3 mL of 3% and one in 3 mL of 12% $H_2O_2$ for 120 minutes each to serve as negative controls. One BI was not exposed to $H_2O_2$ to serve as a positive control.

The BIs were handled using sterile technique and after testing were shipped in sealed plastic bags to STERIS Laboratories. Sterility testing was performed according to STERIS Laboratories protocol ST/01, in which the BIs were incubated in 10 mL of Tryptic Soy Broth at 55–60 degrees Celsius for seven days. The cultures were evaluated daily and interpreted as "No growth" only if the medium remained clear without color change or turbidity after seven days [24].

## Results

### Concentration testing

The concentration strips for all decontamination cycles showed internal concentrations of $H_2O_2$ exceeding 5000 ppm. The exterior concentration strips monitoring for vented $H_2O_2$ all showed undetectable levels of $H_2O_2$ (Tables 2 and 3).

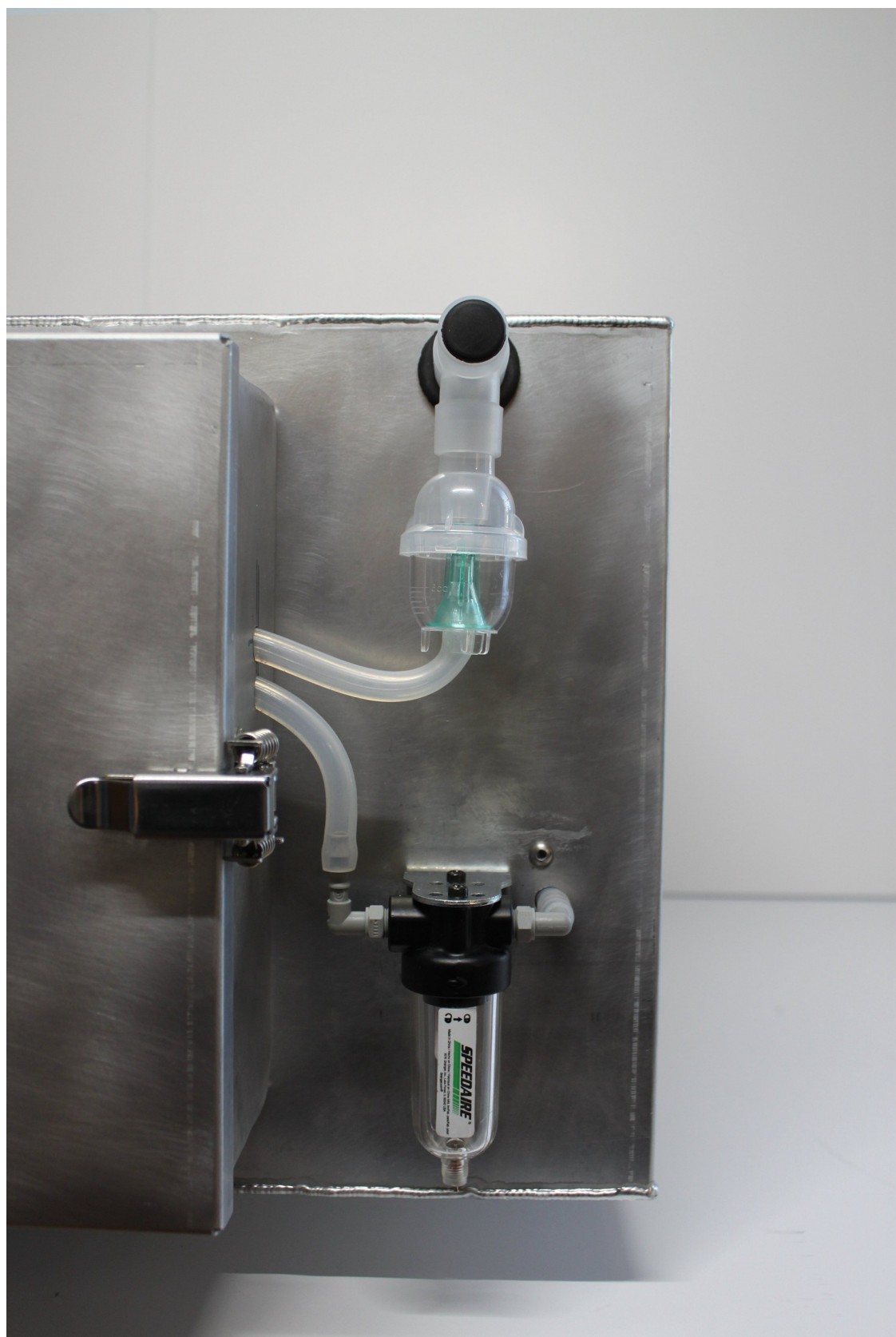

**Fig 3. Arca exterior showing nebulizer (top), condenser unit (bottom), and electronics housing (left).**

**Table 2. Concentration strip results for 3% $H_2O_2$ solution testing.**

|  | 45 Minutes Cycle | 60 Minutes Cycle | 90 Minutes Cycle | 120 Minutes Cycle |
|---|---|---|---|---|
| Internal Strip | >5000 ppm | >5000 ppm | >5000 ppm | >5000 ppm |
| Exterior Strip | No color change | No color change | No color change | No color change |

**Table 3. Concentration strip results for 12% $H_2O_2$ solution testing.**

|  | 45 Minutes Cycle | 60 Minutes Cycle | 90 Minutes Cycle | 120 Minutes Cycle |
|---|---|---|---|---|
| Internal Strip | >5000 ppm | >5000 ppm | >5000 ppm | >5000 ppm |
| Exterior Strip | No color change | No color change | No color change | No color change |

**Table 4. BI growth at 7 days following exposure to 3% $H_2O_2$ solution.**

| Trial | 120 Minutes Submersion | 45 Minutes Cycle | 60 Minutes Cycle | 90 Minutes Cycle | 120 Minutes Cycle |
|---|---|---|---|---|---|
| Growth Result | No growth | No growth | No growth | No growth | No growth |

**Table 5. BI growth at 7 days following exposure to 12% $H_2O_2$ solution.**

| Trial | 120 Minutes Submersion | 45 Minutes Cycle | 60 Minutes Cycle | 90 Minutes Cycle | 120 Minutes Cycle |
|---|---|---|---|---|---|
| Growth Result | No growth | No growth | No growth | No growth | No growth |

## Decontamination testing

All BIs exposed to $H_2O_2$ had no growth at seven days of incubation (Tables 4 and 5). The positive control that was not exposed to $H_2O_2$ demonstrated bacterial growth.

## Discussion

The limited supply of FFRs during the SARS-Cov-2 pandemic endangers health care and other frontline workers. While the FDA and CDC have encouraged the decontamination of FFRs for reuse, existing decontamination systems are inaccessible in low-resource settings. Development of systems designed for deployment in low-resource settings will increase the supply of PPE for at-risk personnel.

This study attempted to evaluate a novel device's ability to eradicate bacterial pathogens from the surface of an FFR without exposing users to unsafe levels of $H_2O_2$. Our testing demonstrates that the novel Arca device can decontaminate bacterial pathogens without exposing users to high concentrations of $H_2O_2$. These results are consistent with the literature on existing gaseous $H_2O_2$ technologies. The eradication of bacteria achieved at every trial suggests that the Arca could achieve adequate decontamination with a lower concentration $H_2O_2$ solution, a shorter cycle duration, or a smaller volume of $H_2O_2$. Based on previously reported $H_2O_2$ testing, the bactericidal properties of the device suggest virucidal efficacy as well [9,25].

The Arca has a throughput of 16 FFRs per cycle, with a daily output of up to 512 FFRs. The marginal cost to clean each additional FFR is significantly less than $1, as operating expenses include small volumes of widely available concentrations of $H_2O_2$ and the variable cost of electricity to power the device. The Arca's production cost at less than $2,000 per unit is significantly lower than the price of existing commercially available solutions. At scale, this manufacturing cost should decrease. These devices are intended to be donated at no cost to

recipient institutions, therefore the operating expenses are the only costs passed on to end users. Future iterations of the device could reduce costs further by using more off-the-shelf components in its manufacturing process. The decreased cost of the Arca improves its accessibility in low-resource settings compared with existing FFR decontamination systems.

## Limitations

The sterility testing performed on the BI strips resulted in a binary "No growth" or "Growth" outcome, and did not demonstrate log kill rates. Therefore, it is possible that small populations of bacteria survived decontamination cycles but did not change the TSB medium's turbidity. However, the same testing methodology is used as part of quality assurance testing to assess sterility achieved by FDA-approved devices, with an industry standard sterility assurance level (SAL) of $10^{-6}$. The FDA requires only an SAL of $10^{-3}$ for products that contact intact skin [26].

Testing was performed under optimal conditions using a single FFR. Real world use may lead to decreased efficacy due to variations in power supply to the Arca, increased pathogen burden, or greater surface area of multiple FFRs. Additionally, while bacteria are considered more resistant to $H_2O_2$ than viruses, the eradication of this study's BIs may not be generalizable to all mutations of SARS-CoV-2.

## Future directions

Experimental derivation of the absolute minimum cycle duration, volume of $H_2O_2$, and percent $H_2O_2$ solution to achieve decontamination will allow for greater throughput of the Arca device. Testing in a BSL-3 lab will evaluate the Arca's efficacy at decontamination of SARS-CoV-2 and other pathogens directly inoculated on PPE.

Fitment testing of FFRs after decontamination will allow for experimental derivation of the maximum decontamination cycles per FFR. Additional testing of other PPE and various equipment may expand the use cases for the Arca.

Additionally, "real world" field testing will help to determine any potential causes of diminished aerosol concentration, inadequate decontamination, or non-optimal user experience.

## Conclusion

The Arca Aerosolized Hydrogen Peroxide Decontamination System can safely decontaminate FFRs inoculated with bacteria using a minimum cycle time of 45 minutes with 6 mL of $H_2O_2$ solution.

## Acknowledgments

We would like to acknowledge the tireless work of our systems engineers, Garry Osborne and Keith Crowder, as well as our head of business development, Michael Mazza. Without their constant support, the Arca would have been neither built nor tested. Our acknowledgements would not be complete without thanking Dr. Sarah Huepenbecker and Natalie Punchak for their sage advice and emotional support.

## Author Contributions

**Conceptualization:** Loren Benjamin Mead, Tanner Mathison.

**Data curation:** Tanner Mathison, Garry Osborne, Anne Marie Richards.

**Funding acquisition:** Tanner Mathison.

**Investigation:** Loren Benjamin Mead, Tanner Mathison.

**Methodology:** Loren Benjamin Mead, Tanner Mathison, Garry Osborne, Anne Marie Richards.

**Project administration:** Loren Benjamin Mead, Tanner Mathison.

**Supervision:** Loren Benjamin Mead, Tanner Mathison.

**Validation:** Loren Benjamin Mead.

**Writing – original draft:** Loren Benjamin Mead.

**Writing – review & editing:** Tanner Mathison, Anne Marie Richards.

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
