## [Decision Letter · Decision Letter 0]

16 Jun 2022

PONE-D-22-12807Decontamination of Geobacillus Stearothermophilus Using the Arca Aerosolized Hydrogen Peroxide Decontamination SystemPLOS ONE

Dear Dr. Loren,

Thank you for submitting your manuscript to PLOS ONE. After careful consideration, we feel that it has merit but does not fully meet PLOS ONE’s publication criteria as it currently stands. Therefore, we invite you to submit a revised version of the manuscript that addresses the points raised during the review process.

We look forward to receiving your revised manuscript.

Kind regards,

Peter Setlow

Academic Editor

PLOS ONE

Journal Requirements:

[The authors of this publication are members of Abaton's Board of Directors, which could be perceived as a competing interest. However, the authors do not receive a salary or any gifts-in-kind for their service on the Board, and Abaton is a 501(c)3 non profit. The Arca device design has not, nor ever will be submitted for patent. The intention of the authors is to disseminate the design as an open source document for replication in locations where it might be of assistance. The authors further declare that they in no way financially or otherwise benefitted from the grant funding and personal donations made to Abaton.]

Additional Editor Comments:

Dear Dr. Loren,

Your manuscript has been reviewed by two experts in the field, and while both reviewers are positive about the manuscript, they make a few minor comments that should be attended to, as they will improve the manuscript.

Sincerely,

Peter Setlow

Reviewers' comments:

Reviewer's Responses to Questions

**Comments to the Author**

1. Is the manuscript technically sound, and do the data support the conclusions?

Reviewer #1: Partly

Reviewer #2: Yes

2. Has the statistical analysis been performed appropriately and rigorously? 

Reviewer #1: Yes

Reviewer #2: N/A

3. Have the authors made all data underlying the findings in their manuscript fully available?

Reviewer #1: Yes

Reviewer #2: Yes

4. Is the manuscript presented in an intelligible fashion and written in standard English?

Reviewer #1: Yes

Reviewer #2: Yes

5. Review Comments to the Author

Reviewer #1: The manuscript by Mead et al "Decontamination of Geobacillus Stearothermophilus using the Arca Aerosolized Hydrogen Peroxide decontamination system" is a well written description of a device that uses vaporized hydrogen peroxide to disinfect N95 respirators. The methods are straightforward and are consistent with previous literature that vaporized H2O2 is an effective disinfectant.

The main issue to consider is that the authors repeatedly discuss this as an option for low resource areas at a cost of 2000. It is not clear from figure 2 how many masks fit in the device and what the throughput is. In table 1 many of those were high throughput set ups such that it is not an equitable comparison. Perhaps including the masks in Figure 2 would be helpful. A cost per N-95 might be more applicable as if one can only repurpose 20 per day even at 2000 it may not be cost effective especially as manufacturing and stockpiles have increased. It would be helpful in the discussion to discuss throughput (how many masks per day for a single unit) as well as cost in this context.

Minor point

The authors state sterility was demonstrated by clear TSB and lack of color change at 7 days. Unfortunately I was unable to access the cited reference (error code) for the steris procedure and the manufacturer of the BI strip is not provided. TSB turbidity requires significant growth and lack of turbidity is not equivalent to no growth. The media will not necessarily change color with small amounts of growth (perhaps this was a reference to a color indicator on the strip) Filtering the media and plating the filter to catch any organisms is one mechanism to capture small amount of organisms in a volume too large to plate. Some clarification of this procedure as well as the limit of detection for colony forming units would be helpful to the reader. It would also be helpful to note how much growth was seen with the positive control as well to confirm the log killing and disinfection achieved if we assume the TSB was in fact sterile.

Reviewer #2: A very straightforward paper. Only a few minor comments.

1) l 42 - In light of ramping up a variety of responses to the pandemic that drove up supplies of PPE and drove down costs, might be helpful to give an estimate of the times/costs to process 1, 10, 100 etc M-95s would be helpful.

2) l-90 . Presumably these are G. stearothermophilus spores? Are these BIs the ones used for H2O2 sterilizers, as I believe these spores are not generally the ones most resistant to H2O2, although are for wet heat.

3) Since the TSB incubated with H2O2-treated BIs gave no obvious turbidity/color change, most spores have been killed. But what are the limits of this assessment? I could not access reference 23 to get any information about this, and thus it is not clear what the confidence in "sterility is.

6. PLOS authors have the option to publish the peer review history of their article (what does this mean?). If published, this will include your full peer review and any attached files.

Reviewer #1: No

Reviewer #2: No

---

## [Author Response · Author response to Decision Letter 0]

16 Aug 2022

Reviewer #1: The manuscript by Mead et al "Decontamination of Geobacillus Stearothermophilus using the Arca Aerosolized Hydrogen Peroxide decontamination system" is a well written description of a device that uses vaporized hydrogen peroxide to disinfect N95 respirators. The methods are straightforward and are consistent with previous literature that vaporized H2O2 is an effective disinfectant. The main issue to consider is that the authors repeatedly discuss this as an option for low resource areas at a cost of 2000. It is not clear from figure 2 how many masks fit in the device and what the throughput is. In table 1 many of those were high throughput set ups such that it is not an equitable comparison. Perhaps including the masks in Figure 2 would be helpful. A cost per N-95 might be more applicable as if one can only repurpose 20 per day even at 2000 it may not be cost effective especially as manufacturing and stockpiles have increased. It would be helpful in the discussion to discuss throughput (how many masks per day for a single unit) as well as cost in this context. 

Response: The comment from Reviewer #1 regarding equitable comparisons of the devices is well-taken. The throughput rates for the commercially available systems are much larger than that of the Arca. However, while the marginal cost to clean an additional N95 is significantly lower for the large throughput commercially available options, the outsized cost of these solutions make them a non-starter for low-resource settings. Additionally, Abaton is a non-profit which provides the Arca devices for free to at-need institutions. The true costs to end users include the purchase of additional H2O2, electricity to run the Arca, and staffing. These are variable costs based on location and utility pricing. We have highlighted this point and included the throughput of the Arca in our updated discussion to better address the cost-effectiveness of the device. 

 Minor point The authors state sterility was demonstrated by clear TSB and lack of color change at 7 days. Unfortunately I was unable to access the cited reference (error code) for the steris procedure and the manufacturer of the BI strip is not provided. TSB turbidity requires significant growth and lack of turbidity is not equivalent to no growth. The media will not necessarily change color with small amounts of growth (perhaps this was a reference to a color indicator on the strip) Filtering the media and plating the filter to catch any organisms is one mechanism to capture small amount of organisms in a volume too large to plate. Some clarification of this procedure as well as the limit of detection for colony forming units would be helpful to the reader. It would also be helpful to note how much growth was seen with the positive control as well to confirm the log killing and disinfection achieved if we assume the TSB was in fact sterile.

Response: This is a valid critique and therefore we have added to our discussion regarding the limitations of this study. The TSB growth medium testing provides a binary outcome of growth versus no growth. STERIS labs were unable to provide log kill information. 

Reviewer #2: A very straightforward paper. Only a few minor comments.  1) l 42 - In light of ramping up a variety of responses to the pandemic that drove up supplies of PPE and drove down costs, might be helpful to give an estimate of the times/costs to process 1, 10, 100 etc M-95s would be helpful. 

Response: We have added the throughput of the Arca in our discussion to better evaluate the cost-effectiveness of the device.

 2) l-90 . Presumably these are G. stearothermophilus spores? Are these BIs the ones used for H2O2 sterilizers, as I believe these spores are not generally the ones most resistant to H2O2, although are for wet heat.

Response: That is correct, we used G. stearothermophilus spores. According to CDC recommendations, G. stearothermophilus is the appropriate biologic indicator for assessing sterilization with H2O2 devices. Please see the link below: https://www.cdc.gov/infectioncontrol/guidelines/disinfection/sterilization/sterilizing-practices.html

 

3) Since the TSB incubated with H2O2-treated BIs gave no obvious turbidity/color change, most spores have been killed. But what are the limits of this assessment? I could not access reference 23 to get any information about this, and thus it is not clear what the confidence in "sterility is.

Response: This is a valid critique and therefore we have added to our discussion regarding the limitations of this study. The TSB growth medium testing provides a binary outcome of growth versus no growth. STERIS labs were unable to provide log kill information.

---

## [Decision Letter · Decision Letter 1]

18 Aug 2022

Decontamination of Geobacillus Stearothermophilus Using the Arca Aerosolized Hydrogen Peroxide Decontamination System

PONE-D-22-12807R1

Dear Dr. Mead,

We’re pleased to inform you that your manuscript has been judged scientifically suitable for publication and will be formally accepted for publication once it meets all outstanding technical requirements.

Kind regards,

Peter Setlow

Academic Editor

PLOS ONE

Additional Editor Comments (optional):

All comments made in the previous review have been answered appropriately, and the data in the manuscript have been presented objectively and in an unbiased manner, and the latter is also true of all conclusions made in the manuscript.

Reviewers' comments:

Reviewer's Responses to Questions

**Comments to the Author**

1. If the authors have adequately addressed your comments raised in a previous round of review and you feel that this manuscript is now acceptable for publication, you may indicate that here to bypass the “Comments to the Author” section, enter your conflict of interest statement in the “Confidential to Editor” section, and submit your "Accept" recommendation.

Reviewer #2: All comments have been addressed

2. Is the manuscript technically sound, and do the data support the conclusions?

Reviewer #2: Yes

3. Has the statistical analysis been performed appropriately and rigorously? 

Reviewer #2: Yes

4. Have the authors made all data underlying the findings in their manuscript fully available?

Reviewer #2: Yes

5. Is the manuscript presented in an intelligible fashion and written in standard English?

Reviewer #2: Yes

6. Review Comments to the Author

Reviewer #2: (No Response)

7. PLOS authors have the option to publish the peer review history of their article (what does this mean?). If published, this will include your full peer review and any attached files.

Reviewer #2: No

---

## [Editor Report · Acceptance letter]

1 Sep 2022

PONE-D-22-12807R1 

Decontamination of *Geobacillus Stearothermophilus* Using the Arca Aerosolized Hydrogen Peroxide Decontamination System 

Dear Dr. Mead:

I'm pleased to inform you that your manuscript has been deemed suitable for publication in PLOS ONE. Congratulations! Your manuscript is now with our production department. 

Kind regards, 

on behalf of

Dr. Peter Setlow 

Academic Editor

PLOS ONE